# FLASH: Flow-Based Language-Annotated Grasp Synthesis for Dexterous Hands

**Abstract:** We introduce FLASH, a method for language-conditioned dexterous grasping that jointly models task intent and physical contact quality for robot hands. Unlike prior approaches, our text-conditioned grasp synthesis pipeline is explicitly aware of geometric information during generation. FLASH learns a single flow-matching model conditioned on hand and object point clouds and natural language instructions. Our model operates on live-updated, vectorized hand meshes and is trained on our improved grasp dataset, FLASH-Drive, which includes refined grasps, water-tight meshes and augmented text annotations. This enables FLASH to outperform prior work in producing physically plausible grasps that align with goals specified via text. We use a pre-trained large language model as the backbone of our architecture, enabling generalization to novel prompts and objects.

**Keywords:** Dexterous Grasping, Flow Matching, Large Language Models

## 1 Introduction

Dexterous robotic grasping has advanced rapidly through differentiable simulation, large-scale datasets, and pre-trained generative models [1, 2, 3, 4]. Yet, most pipelines still decouple *physical plausibility* from *task intent*: geometry-focused methods optimize contact without semantics, while language-conditioned ones sample then refine grasps [5, 6, 7, 8]. This two-stage setup prevents end-to-end credit assignment from physical failures back to language conditioning.

We propose FLASH, a conditional flow-matching network that jointly models language, geometry, and contact dynamics to produce stable, semantically aligned multi-fingered grasps. Unlike prior work that predicts from static hand parameters and refines post-hoc, FLASH re-meshes the evolving hand at every step to feed a live hand point cloud back into the model, using an efficient batched mesh processing pipeline that makes this geometry-aware feedback loop practical at training and inference scale. Our approach builds on recent datasets that pair diverse grasps with text annotations generated from contact data or images [9, 3, 10, 2, 11, 4].

Generative grasp models have used autoencoders for compact latent spaces, transformers for multi-modal fusion, and diffusion models for diverse high-quality samples. Flow-matching offers faster inference than diffusion while preserving quality [12, 13, 14].

Our framework takes language prompts and hand-object point clouds as input, using GPU-accelerated, vectorized meshes regenerated at inference for realism. We also post-process existing datasets to improve physical feasibility while preserving semantic intent. We utilize a pre-trained large language model [15] to introduce broad world knowledge, enabling generalization to novel prompts and geometries.

In summary, this paper makes the following main contributions:

1. FLASH: a flow-matching architecture that generates contact-quality dexterous grasps aligned with language commands.
2. FLASH-Drive: a large-scale, high-fidelity, language-annotated grasp dataset with semantic point clouds, plus code and pre-trained weights.

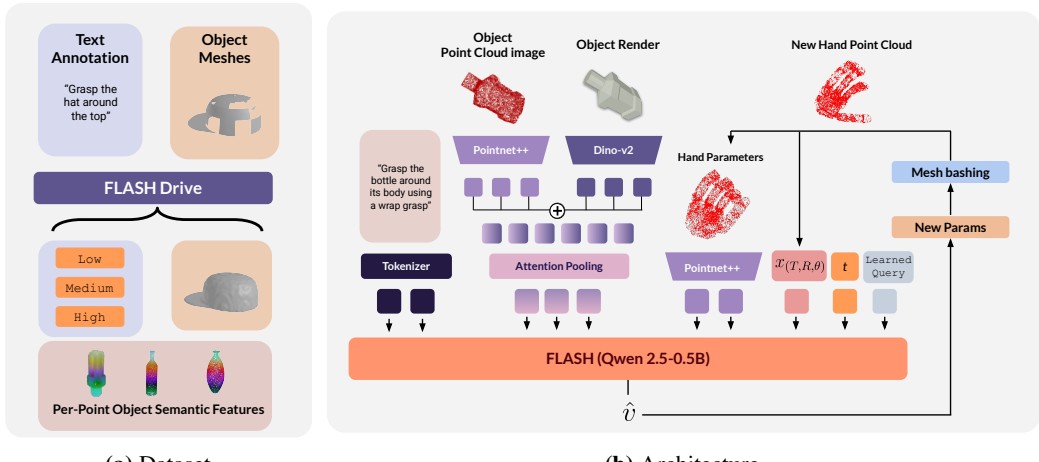

**(a)** Dataset             **(b)** Architecture

**Figure 1:** FLASH is a conditional flow-matching model capable of generating semantically-aligned grasps that are physically plausible. We first produce a dataset by improving the quality of object meshes, improving the physical feasibility of grasps via SDF-based optimization, adding synthetic text annotations, and generating per-point semantic object features using DINOv2. We then train FLASH on this data with a flow-matching objective, incorporating a live updating hand point cloud for geometric awareness and per-point object features to enable semantic generalization.

3. A mesh processing pipeline that enhances existing datasets and enables geometric awareness during flow-matching.

## 2 Method

We present FLASH, a conditional flow-matching approach for generating dexterous robotic grasps from natural language. Trained on FLASH-Drive—our dataset of refined grasps and enriched annotations—FLASH combines dataset refinement, flow-based model design, and an efficient inference pipeline.

### 2.1 Dataset Refinement and FLASH-Drive

We created FLASH-Drive by enhancing the MultiGraspLLM dataset [4] to be both physically plausible and semantically rich. For semantics, we used the OpenAI o4-mini vision-language model with structured JSON outputs to generate over $200\,000$ detailed contact-pattern descriptions from rendered grasps and per-finger contact labels, capturing nuanced functional intent. For physical quality, we built watertight object meshes via voxelization, multi-view depth-based hole filling, and Laplace smoothing, enabling accurate SDF computation. We then refined grasp poses by batch-optimizing hand point clouds ($N_{refine} = 8192$) against object SDFs using PyTorch3D and Kaolin kernels, penalizing object penetrations, large surface distances, and fingertip self-collisions, with independent optimization per hand. This process produced a large-scale dataset of refined grasps, high-fidelity meshes, and diverse language annotations forming the foundation for training FLASH.

### 2.2 Flow-Based Grasp Generation

FLASH learns a single-stage, end-to-end generative process for grasp synthesis, mapping conditioning information directly to stable, semantically relevant grasps. It achieves this using a conditional flow-matching (CFM) framework [14]. The core task is to learn a velocity field $\hat{v}(x_t, t, c)$ that guides the evolution of hand parameters $x_t \in \mathbb{R}^{25} = [T, R, \theta]$, where $T \in \mathbb{R}^3$ is the wrist position, $R \in \mathbb{R}^6$ is the wrist rotation, and $\theta \in \mathbb{R}^{16}$ represents the joint angles of each finger link in radians. $x$ is predicted over a normalized time $t \in [0, 1]$, conditioned on $c$.

Training minimizes the discrepancy between the predicted velocity $\hat{v}$ and the ground-truth velocity $u = x_T - x_0$ derived from grasp pairs $(x_0, x_T)$ sampled from FLASH-Drive. The CFM objective,

applied at randomly sampled times $t$ and interpolated states $x_t = x_0 + tu$, is given by:

$$\mathcal{L}_{\text{CFM}}(\theta) = \mathbb{E}_{(x_0, x_T, c) \sim D} \, \mathbb{E}_{t \sim U[0,1]} \| f_\theta(x_t, t, c) \, - \, (x_T - x_0) \|_2^2, \tag{1}$$

where $f_\theta$ is the neural network parameterized by $\theta$, and $D$ is the FLASH-Drive dataset. To further enforce physical realism during generation, this objective is augmented with a penetration penalty.

$\mathcal{L}_{\text{CFM}}$ constitutes the entire loss for our model. This keeps our objective simple, while the nature of our data generation process allows our model to implicitly optimize for physical plausibility. In this way, we avoid the complexity of weighting several loss terms in order to make trade-offs between physical plausibility and semantic alignment with text conditioning.

The velocity prediction $\hat{v}(x_t, t, c)$ is conditioned on a rich set of inputs represented by $c$. Semantic intent is provided by the natural language prompt, tokenized via the unmodified tokenizer of a pre-trained Qwen-2.5 LLM to leverage its learned representations. Object shape information enters through per-point geometric features $f_{O,geom}$ extracted from the object point cloud $P_O \in \mathbb{R}^{N \times 3}$ by a PointNet++ encoder [16]. To enhance generalization to unseen objects, these geometric features are augmented with semantic context; we extract dense visual features using DINOv2 [17] from renderings of the textured object mesh, project them onto $P_O$, concatenate them with $f_{O,geom}$, and process them through a transformer decoder attending to learned queries. This yields a compact sequence of combined semantic and geometric object features $f_{O,sem} \in \mathbb{R}^{L \times D_{sem}}$. Crucially, the model also receives information about the hand's current geometric state during the flow trajectory. This is achieved by generating the hand mesh corresponding to $x_t$ on-the-fly using an efficient batch meshing pipeline, sampling a hand point cloud $P_H(x_t) \in \mathbb{R}^{M \times 3}$, and extracting its geometric features $f_H \in \mathbb{R}^{M \times D_{geom}}$ via PointNet++. This live hand geometry input allows the model to reason explicitly about potential collisions and contact points throughout generation, overcoming the information bottleneck associated with relying solely on the parameter vector $x_t$. For ablations on semantic features and dynamic updates to the hand point cloud, please refer to Section 3.2.

These conditioning inputs—text embeddings, object features $f_{O,sem}$, live hand features $f_H$, and time $t$—are processed by our network, which employs the Qwen-2.5 LLM architecture as its backbone, ultimately predicting the 25-dimensional velocity vector $\hat{v}$ (see Figure 1b)

## 2.3 Inference

To synthesize a grasp at inference time, given a text prompt and an object point cloud (providing conditioning $c$), FLASH starts from a predefined initial hand state $x_0$. It then simulates the learned dynamics by numerically integrating the predicted velocity field $\hat{v}(x_t, t, c)$ from $t = 0$ to $t = 1$, using an Dopri9 ODE solver. At each integration step, the model's prediction $\hat{v}$ is conditioned not only on the static inputs but also on the live hand geometry $P_H(x_t)$ derived from the hand state $x_t$ estimated for that step. This continuous feedback loop ensures the generated trajectory remains geometrically aware. The resulting state at $t = 1$, $x_T$, is the final predicted grasp configuration $x^*$.

## 3 Experiments

### 3.1 Experimental Setup

**Dataset and Refinement** – All models are trained and evaluated on FLASH-Drive, our enhanced version of the MultiGraspLLM dataset [4]. As detailed in Section 2.1, FLASH-Drive addresses limitations in original mesh quality by providing high-fidelity watertight object meshes. Using these meshes and our efficient vectorized mesh processing pipeline, we refined the original grasp poses via SDF-based optimization to significantly reduce penetration and improve physical plausibility while preserving functional intent. Furthermore, we augmented the dataset by generating over $200\,000$ additional structured text annotations using OpenAI's o4-mini model, describing grasps at low, mid, and high levels of abstraction to improve semantic understanding and generalization. This resulted in FLASH-Drive, a large-scale dataset featuring refined grasps across multiple hand embodiments, paired with rich textual descriptions. Our refinement process demonstrably reduced grasp penetration and improved simulated success rates compared to the original dataset grasps, establishing a higher quality foundation for training.

**Table 1:** Simulation results on **seen** objects (metrics aggregated over test set). Lower is better for CD and Max Pen Dist.; higher is better for Succ. Rate and GPT Score.

| Method | Chamfer Dist.↓ | Max Pen Dist.↓ | Succ. Rate↑ (%) | GPT Score (Align/Feas.)↑ |
|---|---|---|---|---|
| MultiGraspLLM [4] | **0.37** | 1.04 | **31.98** | – / – |
| DexGraspNet [2] | 0.62 | 1.27 | – | – / – |
| **FLASH** (ours) | 0.43 | **0.36*** | 31.34 | **55.2 / 79.0**** |

*Max Pen Dist. for FLASH measured on generated grasps, may differ slightly from dataset refinement target.
**GPT Scores are dataset averages from FLASH-Drive annotations, indicative of model target.

**Table 2:** Ablation study results on seen objects. Lower SDF Loss and CD indicate better geometric quality.

| Variant | SDF Loss (↓) | Chamfer Dist. (CD) (↓) |
|---|---|---|
| Full FLASH | 0.43 | 0.36 |
| w/o Live Hand PC | 0.66 | 0.44 |
| w/o LLM (transformer from scratch) | 0.37 | 0.35 |
| w/o Refinement (Train on Orig. Data) | 0.64 | 0.35 |
| w/o Semantics (DINOv2 features) | 0.57 | 0.31 |
| w/o Lang (No text conditioning) | 0.49 | 0.35 |

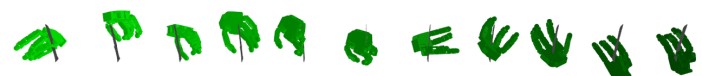

**Figure 2:** A sample flow trajectory from FLASH when prompted with "grasp the knife with all fingers"

**Baselines** – We compare FLASH against key prior works relevant to language-conditioned dexterous grasping. MultiGraspLLM [4], the source of our initial dataset, serves as a primary baseline representing recent LLM-based approaches. Additionally, we compare against DexGraspNet [2], a large-scale generative model, representing geometry-centric methods; comparisons are based on geometric metrics evaluated on its generated grasps for comparable objects.

### 3.2 Ablation Studies

We conducted ablations on the seen object set (Table 2) to assess FLASH 's design choices. Removing live hand point cloud feedback (w/o Live Hand PC) greatly worsened geometric quality (higher SDF Loss, CD), confirming the need for continuous geometric reasoning. Replacing Qwen-2.5 with a scratch-trained transformer (w/o LLM) mainly hurt semantic understanding. Training on unrefined grasps (w/o Refinement) degraded geometry, validating FLASH-Drive's refinement. Dropping DINOv2 semantic features (w/o Semantics) reduced geometric performance, showing their contextual value. Omitting language conditioning (w/o Lang) also hurt geometry, indicating language guidance helps constrain generation toward valid, semantically aligned grasps.

## 4  Conclusion

In this work we present FLASH, a text-conditioned, geometry aware, robot grasping model. Alongside it, we also release FLASH-Drive, a multi-embodiment robot grasping dataset with contact annotations of varying levels of details and improved contact quality. By feeding back a reconstructed hand point cloud back to FLASH as its flowing the hand parameters, we are able to generate higher quality grasps and generalize to new object geometries while not preserving quick inference speed.

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

## A    Implementation and Metric Details

**Metrics** – We evaluate performance using several metrics assessing both physical validity and semantic correctness. Physical plausibility is measured primarily by the Maximum Penetration Distance (Max Pen Dist., ↓) derived from SDFs (Table 1) and an aggregate SDF Loss (↓) used in ablations (Table 2), where lower values indicate less interpenetration. Geometric similarity to ground truth is assessed using Chamfer Distance (CD, ↓) (Table 1, Table 2). Functional success is quantified by the Simulation Success Rate (Succ. Rate, ↑), the percentage of grasps successfully lifting and holding objects in simulation via a standardized heuristic.

**Simulation Evaluation Heuristic** – Our grasp evaluation approach leverages the IsaacLab simulator, employing a heuristic similar to that of DexGraspNet [2]. This procedure begins with the hand in a pre-grasp pose (flat hand), then moves the hand towards the object based on predicted wrist pose $(T, R)$. Subsequently, a motion plan brings the hand joints towards the target configuration $\theta$. Finally, the hand attempts to lift the object vertically. Grasp success is determined by checking if the object is lifted above a threshold height and remains stable (minimal velocity) after a short duration. This evaluation heuristic allows for scalable testing of numerous grasp candidates efficiently.

**Implementation Details** – We primarily use the Allegro Hand within the IsaacLab simulator for simulation results presented in the main paper, and the LEAP hand for real-world demonstrations. Our FLASH architecture utilizes the Qwen-2.5 (0.5B) LLM backbone and a PointNet++ [16] encoder pre-trained for part segmentation on ShapeNet [18]. Semantic features are derived from DINOv2 [17]. Models were trained on a single NVIDIA H200 GPU for approximately 6 hours.

## B    Real world experiments

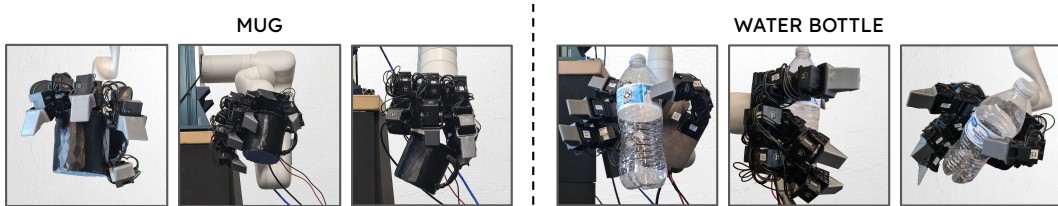

**Figure 3:** Real-world demonstration setup using a LEAP hand mounted on a Kinova Gen 3 arm in a laboratory environment, used for qualitative validation.

### B.1    Real-World Robotic Demonstration

Finally, to assess FLASH's applicability in the real world, we conducted qualitative experiments using a Kinova Gen3 robot arm equipped with a LEAP hand [19], shown in Figure 3. We selected several common household objects (e.g., mug, water bottle, drill) and provided text prompts representing typical functional intents (e.g., "grasp the handle to lift", "pick up the bottle by the body"). Grasps were generated offline by FLASH, assuming access to the object mesh, and then executed open-loop on the robot system by moving to the predicted pose and closing fingers to the predicted joint angles. Qualitatively, FLASH successfully generated functionally appropriate and physically stable grasps across the tested objects and prompts. The robot was observed to securely grasp items according to the instructions, for example, correctly using a power grasp for the drill handle when prompted versus attempting a different grasp if only asked to 'pick up'. While these demonstrations are qualitative and do not involve closed-loop control, they strongly suggest that grasps generated by FLASH can transfer effectively to physical hardware and follow nuanced language instructions in practice. We also observed qualitatively that language conditioning significantly influenced the grasp strategy towards the intended function compared to a non-conditioned variant which often defaulted to more generic grasps.

## C  Limitations

Despite promising results, FLASH has limitations reflecting the challenges in conditional grasp synthesis. First, geometric accuracy and input requirements pose difficulties. Performance degrades on thin structures due to Signed Distance Function (SDF) limitations, and the current reliance on complete object point clouds hinders application with partial sensor data or vision-only inputs.

Second, inference speed is a constraint. Generating grasps via iterative ODE solving is computationally intensive, currently limiting real-time use and involving a trade-off between generation speed and final grasp quality. Third, generalization capabilities are bounded by the training data. While FLASH-Drive is large, its object diversity primarily covers a limited set of roughly 40 semantic categories, potentially restricting generalization to truly novel object types. Similarly, robustness to natural language commands significantly diverging from the structured prompts seen during training requires further investigation.

Finally, the focus on static grasp generation means limited consideration of the broader task context. Robust sim-to-real transfer needs further development beyond current qualitative demonstrations, and factors like post-grasp stability under load or suitability for subsequent manipulation steps are not explicitly modeled. Addressing these challenges—improving geometric handling, accelerating inference, broadening generalization, bridging the sim-to-real gap, and incorporating task context—are key directions for future work.

