# OpenReview forum: "FLASH: Flow-Based Language-Annotated Grasp Synthesis for Dexterous Hands"
_robot-learning.org/CoRL/2025/Workshop/Dexterous_Manipulation — CoRL 2025 Workshop Dexterous Manipulation Spotlight_

### Official Review · Reviewer_7RYt · 2025-09-04
**None**

**Rating:** 7
**Confidence:** 3

**Review:**

This paper presents FLASH, a flow-matching (conditional flow) model that generates dexterous grasp trajectories conditioned on an object point cloud and a natural-language instruction. Key novelties include (1) conditioning on a dynamically reconstructed hand point cloud during trajectory integration (“live hand PC”) to provide geometry-aware feedback, (2) leveraging an LLM encoder for language-conditioned grasp intent, and (3) building an improved dataset FLASH-Drive (mesh repair, SDF refinement, and large-scale language annotations). The model predicts per-step velocity fields on the hand-state manifold and integrates them with an ODE solver; experiments show reduced penetration and qualitative language-aware grasp behaviors, plus ablations suggesting the live hand point-cloud condition materially improves geometric quality.

Strengths
1.The technical idea — combining conditional flow matching with a dynamic hand geometry feedback channel — is novel and well-motivated for generating physically consistent dexterous trajectories.
2.Substantial dataset engineering (mesh repair, SDF-based refinement, and structured language annotations) strengthens the training signal and is likely to be a useful community contribution.
3.Ablations convincingly show that the live hand point-cloud condition helps reduce geometric violations (penetration), which supports the central design choice.
4.The paper is well-written and the pipeline is described in reasonable detail for a conference submission.

Weaknesses
1.Sim-to-real validation is weak / qualitative. The real-robot experiments are open-loop and qualitative; there is no quantitative sim→real evaluation (real success rates, repeatability, failure taxonomy). For grasping work that claims geometry- and instruction-aware generation, stronger real-world validation is important.
2.Dependence on watertight meshes / full SDF and limited perception realism. FLASH relies on repaired/watertight meshes and SDF optimization. It is unclear how the method performs under realistic partial/occluded point clouds, noisy depth, or scanning artifacts — the current evaluation does not sufficiently test these scenarios.
3.Evaluation inconsistencies / metric clarity. Some reported metrics (e.g., Chamfer Distance values in different tables) appear inconsistent; the manuscript does not fully specify Chamfer implementation details (sampling counts, normalization, averaging procedure), nor the exact evaluation pipeline (post-processing of generated grasps). This reduces reproducibility and confuses comparison to baselines.

---

### Official Review · Reviewer_jqEe · 2025-09-10
**Interesting method for language-conditioned grasp generation; weak evaluation**

**Rating:** 6
**Confidence:** 4

**Review:**

The authors propose FLASH, a conditional flow-matching model generating physically plausible dexterous grasps based on language annotations. They also introduce FLASH-Drive, a large scale dataset of language-annotated grasps.

FLASH-Drive is an interesting scientific contribution of its own, as the authors perform several non-obvious processing steps on top of the MultiGraspLLM dataset.

The FLASH method itself uses flow-matching to generate hand and wrist parameters for grasps from point clouds and language prompts.

The method shows performance that is competitive with existing approaches for grasp generation, while additionally showcasing alignment with text prompts. An ablation study also confirms the various design choices for the method.

There are a few weaknesses: The evaluation itself is limited to only seen objects, and there are no baselines showcasing GPT score for alignment. For success rate, the original MultiGraspLLM baselines performs better, and it would be good to see the reason for this further explored. The real world evaluation is limited and only qualitative. I believe that for a stronger submission, more in-depth and generalization evaluations should be performed.

Overall, the authors demostrate that their method is an advancement of the state of the art for grasp synthesis in the language conditioned setting.

---

### Decision · Program_Chairs · 2025-09-18

Accept (Spotlight)